# Study of Membranes with Nanotubes to Enhance Osmosis Desalination Efficiency by Using Machine Learning towards Sustainable Water Management

**DOI:** 10.3390/membranes13010031

**Published:** 2022-12-26

**Authors:** Abdelfattah Amari, Mohammed Hasan Ali, Mustafa Musa Jaber, Velibor Spalevic, Rajko Novicevic

**Affiliations:** 1Department of Chemical Engineering, College of Engineering, King Khalid University, Abha 61411, Saudi Arabia; 2Research Laboratory of Processes, Energetics, Environment and Electrical Systems, National School of Engineers of Gabes, Gabes University, Gabes 6072, Tunisia; 3Computer Techniques Engineering Department, Faculty of Information Technology, Imam Ja’afar Al-Sadiq University, Najaf 10070, Iraq; 4Computer Techniques Engineering Department, Dijlah University College, Baghdad 10070, Iraq; 5Computer Techniques Engineering Department, Al-Farahidi University, Baghdad 10070, Iraq; 6Biotechnical Faculty, University of Montenegro, Mihaila Lalica 1, 81000 Podgorica, Montenegro; 7Faculty of Business Economics and Law, Adriatic University, 85000 Bar, Montenegro

**Keywords:** desalination, polyamide reverse osmosis membrane, carbon nanotubes, artificial neural network, machine learning

## Abstract

Water resources management is one of the most important issues nowadays. The necessity of sustainable management of water resources, as well as finding a solution to the water shortage crisis, is a question of our survival on our planet. One of the most important ways to solve this problem is to use water purification systems for wastewater resources, and one of the most necessary reasons for the research of water desalination systems and their development is the problem related to water scarcity and the crisis in the world that has arisen because of it. The present study employs a carbon nanotube-containing nanocomposite to enhance membrane performance. Additionally, the rise in flow brought on by a reduction in the membrane’s clogging surface was investigated. The filtration of brackish water using synthetic polyamide reverse osmosis nanocomposite membrane, which has an electroconductivity of 4000 Ds/cm, helped the study achieve its goal. In order to improve porosity and hydrophilicity, the modified raw, multi-walled carbon nanotube membrane was implanted using the polymerization process. Every 30 min, the rates of water flow and rejection were evaluated. The study’s findings demonstrated that the membranes have soft hydrophilic surfaces, and by varying concentrations of nanocomposite materials in a prescribed way, the water flux increased up to 30.8 L/m^2^h, which was notable when compared to the water flux of the straightforward polyamide membranes. Our findings revealed that nanocomposite membranes significantly decreased fouling and clogging, and that the rejection rate was greater than 97 percent for all pyrrole-based membranes. Finally, an artificial neural network is utilized to propose a predictive model for predicting flux through membranes. The model benefits hyperparameter tuning, so it has the best performance among all the studied models. The model has a mean absolute error of 1.36% and an R^2^ of 0.98.

## 1. Introduction

One of the most necessary reasons for research on water desalination systems and their development is the problem of water shortage and the resulting crises in the world. More than half of the world is considered to be arid and semi-arid areas; therefore, it is not unreasonable to expect a water crisis in it [1,2,3]. Therefore, the need to renew water resources and find a solution to the water shortage crisis is vital. One of the most important ways to mend this problem is to use water purification systems for wasted water resources [4,5,6,7,8]. Among the purification systems with desalination, reverse osmosis has been used for many years due to the reduction in energy consumption and high efficiency. At first, reverse osmosis was used to desalinate sea and salt water [9,10].

The lack of water resources, qualitatively and quantitatively, is considered a serious threat to the world’s population, especially developing countries. 13% of the world’s population still does not have access to safe water sources [11]. However, the common methods of water purification are not the answer to the existing problems. This research tries to solve this problem with nanotechnology methods and compounds. In recent years, nanoparticles have been researched in various areas, including multiphase flows [12,13,14,15] and catalysts [16]. Nanofluids are extensively studied to enhance heat transfer capabilities [17,18]. However, recently, nanoparticles have been developed for water purification worldwide. Water purification is divided into filtration and disinfection, and nanotechnology is used in both parts [19,20]. Nanotechnology provides many nanomaterials for the treatment of surface water, underground water, and wastewater and the removal of toxic metal and organic, inorganic pollution and microorganisms. This research solves the problems in the field of water purification based on various nanomaterials. Using nanofiltration, natural organic substances, microbial and organic pollution, nitrate, and arsenic can be removed from the surface and underground water [21]. Utilizing reverse osmosis, organic and inorganic compounds and microbial contamination can be removed from the water and desalinated [22]. Nanomaterials can eliminate the contamination of toxic metal ions, inorganic and organic compounds, and microorganisms in surface water, underground water, and wastewater [23]. With the help of nano adsorbents, catalysts, and nanomembranes, microbial and chemical pollution of water can be removed [24]. Many research studies are related to the utilization of nanomembranes in reverse osmosis desalination [25,26,27,28,29,30]. A thorough review of the methods is presented in the following.

The increased industrial demands to conserve water, reduce energy consumption, control corrosion, and recycle useful materials from waste streams led to the creation of new and economic applications for these membranes [31,32,33,34,35,36]. In addition, the advancement of biotechnology and pharmaceutical knowledge, along with the development of membrane use, made the method of using membranes a vital step in the separation operation in order to save energy and prevent the heat loss of products, preferably to the distillation method [37]. Scientific research on membrane principles was established by FILMTEC with the production of the FILMTECFT30 membrane in 1963 [38].

With the development of reverse osmosis membrane technology, many changes have been made to optimize membrane performance and energy [39,40]. After the construction of asymmetric membranes, thin-shell composite membranes made significant progress due to the high potential of modification in the structure and optimization of membrane efficiency. Thin shell membranes consist of three layers, including the bottom layer of non-woven fabric, the middle layer of polymers such as polysulfone or polyether sulfone, and the upper layer of thin shell membranes, which are generally made by surface polymerization. The last layer, which is called the polyamide layer, is the most effective layer for salt purification and making a reverse osmosis membrane because it minimizes the holes on the membrane surface to the point where only water molecules can pass through the membrane surface [41,42].

With the development of nanoscience, the water treatment industry has undergone significant changes. Nanoparticles have been used in the absorption of various heavy metals such as lead [43], chromium [44], virus removal from drinking water [45,46], and removal of other pollutants from water. Additionally, in the synthesis of polyamide membranes, nanoparticles have been used to improve the surface. All nanocomposite membranes have had better performance and higher efficiency compared to simple polyamide membranes. Among all the nanocomposites used in membrane synthesis, carbon nanotubes are used in a variety of research [47] because of their special structure and unique features.

In order to optimize the performance of membranes, various materials have been used to modify the nanotubes, and new nanocomposites have been produced, among which silver nanotubes and multiwalled carbon nanotubes can be used to improve the influx by 20% [48]. By using different percentages of HNO_3_ and H_2_SO_4_ acid, Kim et al. [49] created the O-H functional group for more hydrophilicity. Moreover, Barona et al. [50] used single-walled carbon nanotubes along with aluminosilicate for a significant increase in the influx. Additionally, Chan et al. [41] used single-walled carbon nanotubes with zwitterion groups to improve the reverse osmosis polyamide membrane and increase the influx from 12 $\frac{L}{m^{2}h}$ to 32.8 $\frac{L}{m^{2}h}$.

Working for long periods of time, current membranes, and in general, simple polyamide membranes are faced with the clogging phenomenon. This clogging can significantly reduce the membrane influx. Therefore, several methods have been used to prevent the reduction of the influx and the occurrence of concentration polarization on the surface of the membrane [51]. Since the mechanism of reverse osmosis membranes is the diffusion of particles on the membrane surface, the modification of the membrane surface by using materials that can improve the diffusion of water particles and reduce the concentration of fat and salt on the membrane surface has increased. Nanoparticles, especially those with hydrophilic groups on their surface, have been effective in this field. Researchers have shown that membranes with hydrophilic nanocomposites increase the influx and significantly reduce clogging [52].

While using conventional methods to develop the predictive model was common [53,54,55,56,57] in all areas, recently, more research has been devoted to studying the predictive models proposed using machine learning algorithms [58,59,60,61,62,63]. To this end, some studies have been concerned with using artificial neural networks (ANN) [64,65], genetic algorithms, and other machine learning algorithms [66,67,68] in engineering problems. Ruiz-Garcia and Feo-Garcia [69] applied artificial neural networks to a seawater reverse osmosis (SWRO) and estimated the system’s cost. Joy et al. [70] utilized response surface methodology and machine learning algorithms to find the optimized case for the removal of organics from reverse osmosis. The output parameters of their study were the total organic carbon removal and chemical oxygen demand removal. Salgado-Reyna et al. [71] carried out an experiment in a can manufacturing process. They studied the wastewater from this system, and they used the wastewater to desalinate it using reverse osmosis. Through this process, they gathered data to propose models for predicting total dissolved solids and maximum effluent recovery. Their model had a coefficient of determination of more than 0.97.

In this research, a nanocomposite containing carbon nanotubes was made due to the mentioned effects in improving the performance of the membranes with carbon nanotubes. However, because of the increase in the hydrophilicity of the membrane surface, materials with hydrogen bonds were selected. Finally, the use of raw carbon nanotubes coated with pyrrole as a new nanocomposite was investigated to check the performance of the membrane. Additionally, the increase of the influx by these changes, the decrease in the amount of membrane surface clogging in a specific period of time, and the long-term stability of the membrane surface are determined. Finally, using the data of the research, we have proposed predictive models using artificial neural networks to predict the influx of the membrane and the clogging of the membrane.

## 2. Materials and Methods

Multiwalled carbon nanotubes (with a length of about 10 to 20 μm, an outer diameter of 20 to 30 nm, and an inner diameter of 5 to 10 nm) were purchased from CNT Co., Ltd., (Yeonsu-Gu, Incheon, Republic of Korea). 250 mL of pyrrole was purchased from Sigma Aldrich (St. Louis, MO, USA). The polysulfone grains for making the ultrafiltration layer were obtained from BASF, Ludwigshafen, Germany. Other materials used include methylphenylenediamine (MPD), dimethylformamide (DMF), camphor sulfonic acid, polyvinylpyrrolidone (PVP), Sodium dodecyl sulfate powder, triethylamine (TEA), and an inorganic solvent was provided, which is Hexane from Merck, Darmstadt, Germany. In order to perform the clogging test, 250 g of bovine serum albumin (BSA) was purchased from Merck. Hollytex 3329 non-woven fabric was obtained from Merck, Germany, with a thickness of 170 μm. Other materials used for the synthesis of Multiwalled carbon nanotubes, including toluene sulfonic acid and aluminum persulfate, were also obtained from Merck.

### 2.1. Membrane Synthesis

To synthesize the membrane in this research, we first made the ultrafiltration support, and then it was stretched on the non-woven fabric. Then, the membrane was placed in three different baths, including the MPD bath (in the synthesis of the nanocomposite membrane, nanotubes are also added to this bath), the TMC bath, and finally the hexane bath to make the top layer.

The reverse osmosis thin-shell membrane was fabricated in two steps. First, the polysulfone support was formed on the non-woven fabric by the phase separation method, and then a selective polyamide layer was made on it by the interfacial polymerization method by the reaction of two monomers. To make the support, polysulfone powder was first poured into the DMF solution, and then it was stirred in a mixer for 24 h at a temperature of 50 degrees Celsius at a constant speed until it dissolved. Finally, the prepared solution was placed in a dark space to reduce the number of bubbles. The polyester substrate was used for membrane strength. Then, the prepared solution was poured on the same substrate, and a layer of it with a thickness of about 170 microns was stretched on the polyester layer by elastic film, and then it was quickly immersed in the anti-solvent water bath until it solidified [72].

The support prepared for the reverse osmosis membrane was placed in two separate baths containing MPD and TMC, and after combining the MPD solution with distilled water for 10 min, it was poured onto the polyether sulfone membrane, and the rest of the MPD was removed from the membrane by a plastic roller tube. After that, the membrane was saturated with MPD for 2 min in the solution containing 0.15% by the weight of TMC and placed in hexane solvent. After performing these steps and the final washing in hexane, in order to produce a dense polyamide layer, the constructed membrane was kept at 70 degrees Celsius for 10 min. Finally, the prepared membrane was kept in distilled water.

In the next step, a raw carbon nanotube coated with polypyrrole was prepared. For this purpose, 95.15 g of ammonium persulfate was mixed with 375 mL water for 10 min. Then, the combination of toluene sulfonic acid, including 37.13 g of acid and 25 mL of water, was added to the previous solution and stirred for 15 min. After performing the mentioned steps, 5 mL of pyrrole was mixed with water, 0.048 g of multiwalled carbon nanotube was added to the solution, and it was ultrasonicated for 10 min. Then, the solution was slowly stirred into the acid solution for 4 h at room temperature. Next, to separate the black solution from the bottom of the container, a centrifuge was used with a maximum speed of 4200 rpm for 8 min. Finally, the obtained black material was placed in a vacuum oven at a temperature of 60 degrees Celsius for 48 h to dry the solution. After making the nanocomposite for the synthesis of nanocomposite membranes, the modified carbon nanotubes powder was placed in the TMC solution in the first stage of surface polymerization, i.e., in the TMC solution, and the ultrafiltration membrane was made like simple polyamide membranes was placed in this solution for 10 min. The rest of the steps were carried out in the same way as the simple polyamide membrane synthesis. The fabricated membranes were evaluated in influx, desalination, and clogging test parameters.

### 2.2. Membrane Quality Check

To identify the nanoparticles and how they combine and form the polyamide layer on the membrane, SEM analysis was used by the TESCAN measurement device made in the Czech Republic. Based on the identification of surface roughness and the topology on the membrane surface, AFM analysis was used. To calculate the roughness in the membrane analysis, three parameters were used: height difference between the highest edge and the lowest with the $S_{y}$ parameter, the uniformity of the ups and downs with the $S_{q}$ parameter, and the average roughness of the entire membrane surface with $S_{a}$. The device used to evaluate membrane surface roughness was DualScope C-26, DME Corp. Denmark, and DME/AFM software version 2.1.1.2 was used to calculate the mentioned parameters.

FTIR analysis was used to detect the chemical bonds made after the reaction. In this analysis, new compounds of the manufactured composite material were identified, and the existence of bonds formed after the material was modified or combined with new materials was confirmed [73]. In this research, FTIR analysis was used to modify multiwalled carbon nanotubes and the presence of hydroxide groups on the nanotube surface. All FTIR analyses were performed with an ABB-104 Bomem FTIR spectrometer (MB) (ABB Ltd., Zürich, Switzerland).

Contact angle analysis was used to evaluate the hydrophilicity of the manufactured nanocomposites. If the contact angle of water molecules with the membrane surface is lower, the hydrophilicity of the membrane surface is higher. The contact angle measuring device was G10, Kruss, made in Germany.

### 2.3. Membrane Analysis and Test

All fabricated membranes were tested in the device. The test device is made of a pump, feed tank, and tube with an effective surface of 36 ${\mathrm{cm}}^{2}$ (9 cm×4 cm) with the lateral flow, the membrane is placed in the tube, and the water passes through the membrane and returns to the tank. Figure 1 shows the details of this study’s reverse osmosis purification system. According to Figure 1, three cells are placed side by side, and water enters the cells from the feed tank through a pump. The pressure gauge is set at the beginning of each cell, which is about 16 bar in the first cell, 15 bar in the second cell, and about 14 bar in the third cell. The average pressure is calculated and assumed constant for all results for influx calculations.

The amount of water passing through the membrane enters the container for measuring water volume through the tube at the end of it, and the water permeability is determined by Equation (1) [74].
(1)$$J=\frac{V_{p}}{A\times t}$$

In this equation, *J* is the passing flux of water in terms of $L/m^{2}h$, *V_p_* is the volume of water passing through the membrane in liters, *A* is the effective surface of the membrane in square meters, and *t* is the time of the passing water in a certain measured volume in hours.

To determine the amount of salt removal, an electrical conductivity meter model TES made in Taiwan was used. The salt ion concentration in the tank and in the purified water, after passing through the membrane, is measured by Equation (2) [74].
(2)$$R=\left(1-\frac{c_{p}}{c_{f}}\right)\times 100$$
where $C_{p}$ is the concentration of salt in the purified water, and $C_{f}$ is the concentration in the feed solution in terms of microsiemens/cm.

All experiments were carried out under salt pressure in the feed tank equivalent to the electrical conductivity of 4000 microsiemens/cm. It should be noted that all concentrations were calculated based on microsiemens/cm using an electrical conductivity meter and finally included in the mentioned formulas to calculate the influx and the percentage of salt rejection. The water temperature in the feed tank was kept almost constant. Changes were made only in the type of membrane, and different membranes (simple polymer membrane and modified nanoparticle membrane) with different amounts of salt removal and clogging effects were investigated. The membrane clogging resistance test was done by the same device with BSA protein solution filtration. All the membranes were prepared for the clogging test for 24 h after the influx and rejection tests.

In the present study, raw multiwalled carbon nanotubes coated with pyrrole in four different weight percentages (0.001, 0.002, 0.005, and 0.01) were prepared for the test. To check the performance of the membrane, each of these concentrations was brought with the code R-CNT-P(1,2,3,4), which represents raw carbon nanotubes with pyrrole. Additionally, to show simple polyamide membranes in the diagrams, the Bare-PA code was used. Each test consisted of two stages, and the first test was conducted for 90 min of each membrane to evaluate the amount of water passing through the surface of the membrane and to measure its percentage. At this stage, there was only salt-water solution with a concentration of 4000 microsiemens/cm water-salt in the feed tank. In the second stage of the test, 0.4 g of serum albumin fat was added to the water-salt solution with the same specified concentration. Each clogging test lasted for 24 h, and the water flow rate was measured every hour. 

For each specific weight percentage with a specific carbon nanotube, six similar membranes with an area of 36 ${\mathrm{cm}}^{2}$ were made, although a simple polyamide membrane was made and tested to compare the difference between a simple membrane and a nanocomposite.

## 3. Results and Discussion

This section may be divided by subheadings. It should provide a concise and precise description of the experimental results, their interpretation, as well as the experimental conclusions that can be drawn.

### 3.1. Investigation of Synthesized Membranes for Reverse Osmosis Process

Figure 2 shows the SEM analysis for raw carbon nanotubes with different concentrations of pyrrole. The amine group on the surface of the pyrrole-containing membranes has made them denser, which can be seen in Figure 2.

In the results of [75], it is mentioned that, in raw carbon nanotubes with pyrrole, the surface of the nanotube is softer than without pyrrole, and this is due to the structure of the bonds created by the pyrrole coating, which has caused a significant increase in influx and improved clogging.

Figure 3 shows the FTIR analysis for raw and pyrrole-coated multiwalled carbon nanotubes. For the modified carbon nanotube, two new peaks related to the hydrogen band with wavelengths of 687 (N-H) and 3300 (C-H) were observed, and the presence of the hydrogen group is proof of the modification of the carbon nanotube. These hydrogenated bonds contribute to the hydrophilicity of the carbon nanotube and improve the passage of water through the membrane. Additionally, the peaks of 1385 and 1540 are created due to the type 2 amine functional group on the surface of the nanotube [76].

Contact angle analysis was used to determine the hydrophilicity of membranes. According to Figure 4, the contact angle of water with the membranes with raw nanotubes coated with pyrrole is significantly lower than the rest of the manufactured membranes, and this means that water droplets are absorbed by the raw nanotubes coated with pyrrole. The absorption of water droplets into raw nanotubes along with pyrrole creates a hydrophilic layer on the surface of the membrane and prevents fat from reaching it, which leads to a reduction in clogging. This increase in the hydrophilic surface in the fabricated nanocomposite is due to the presence of hydrogen bonds on the surface of carbon nanotubes. Therefore, the hydrophilicity of modified carbon nanotubes is higher than other polyamide membranes [77].

AFM analysis was performed to check the surface roughness and prove the clogging performance based on the used nanocomposites. This test was performed for three types of membranes in three repetitions, and finally, the average data obtained are presented in Table 1. The surface roughness parameters were carefully checked by DME/AFM software version (2.1.1.2), and among these parameters, the most important parameter that is considered to check the roughness level is the $S_{a}$ parameter. This parameter considers the average roughness of the entire surface, and according to the results, the best membrane in terms of the lowest roughness is the MACNTs-PPy nanocomposite membrane with a concentration of 0.001%, and it also showed the lowest amount of clogging in the clogging reduction graph.

The roughness in the nanocomposite membranes with pyrrole decreased to a suitable ratio, but with the addition of nanoparticle concentration higher than 0.005, the roughness increased again. This increase may be due to the ionic interaction between the new groups added on the surface of the membrane, which in some areas leads to folding and increasing the roughness. Two reasons can be proposed for the change of membrane surface roughness compared to simple polyamide membranes. First, with the addition of N-H groups, the empty spaces on the membrane surface have been filled and made the membrane smoother, and the second reason can be attributed to the migration of carbon nanotubes on the membrane surface after being dipped with membrane-forming materials and making the surface uniform with its particles [78]. Surface topology analysis, according to Figure 5, showed that the roughness has decreased compared to the plain polyamide membrane, and the surface of the membrane has become smoother. In general, the structure is observed in all simple polyamide membranes and nanocomposite membranes.

### 3.2. The Performance of Reverse Osmosis Membrane in Transmittance and Rejection

The transmittance and percentage of salt removal from raw nanotube membranes are given in Figure 6. As can be seen, the water flux passed through the membranes with raw nanotubes higher than the membrane without nanotubes. This phenomenon can be due to the hydrogen bond and the sliding of water molecules from the surface of nanoparticles, as well as the tubular structure of carbon nanotubes and the passage of water through the nanotube channels [79]. As seen in the SEM analysis, the carbon nanotubes made the membrane surface smoother than the membranes without nanoparticles. Therefore, the amount of water passing through these membranes is higher. The amount of water flux increases with the increase in the concentration of nanotubes, so that, at the concentration of 0.001%, the amount of passing water flux is 29.5 $L/m^{2}h$, but at the next concentrations, especially at 0.005%, the flux is 30.8 $L/m^{2}h$. According to Figure 6, the salt rejection in all membranes with pyrrole is higher than the normal membrane, and it is above 95%. According to Figure 6, the highest amount of salt removal is related to raw nanotubes with pyrrole at a concentration of 0.001%, which can be due to the good spread on the surface of the membrane and the way the molecules are placed on it, the functional groups with hydrogen on the surface of the membrane, which has happened well in this membrane [80]. Indeed, other membranes also significantly remove NaCl salt from the feed solution, with a very small difference compared to this membrane. Additionally, compared to the commercially made membranes and according to the method of synthesis of the membrane, it can be said that it has an acceptable performance compared to the materials used.

### 3.3. Membrane Clogging

Membrane clogging is one of the most important parameters of membrane investigation. In general, the lower the clogging, the better the performance of the membrane. Nowadays, various technologies are used to improve membrane performance. The amount of clogging in membranes with nanotechnology has been significantly reduced [81,82]. The clogging of the fabricated nanocomposite membranes and the comparison with the plain membrane are shown in Figure 7. As can be seen in Figure 7, all nanocomposite membranes have a higher flow rate than polyamide membranes within 24 h. It must be mentioned that, in order to gather enough data for the artificial neural networks, the membranes are evaluated every hour, and the data is stored. However, in the following section, the representatives of the mentioned data are presented. Additionally, among all the tested nanocomposite membranes, concentrations of 0.001% and 0.005% have the highest flux even at the end of the 24 h test compared to other membranes. Figure 8 is the normalized graph of all the nanocomposite membranes. According to Figure 8, all the nanocomposite membranes have uniform stability, and during 24 h, they are more stable than the simple polyamide membranes. The most stable membrane in the normalized diagram corresponds to the concentration of 0.001%. Additionally, in the concentration of 0.005%, proper stability and uniform flux were observed for 24 h. This concentration has the highest amount of flux, and it can be said that it has the best performance among all membranes.

## 4. Artificial Neural Networks

In order to propose models for the two main parameters of the present study, the data from the experiments are used. We aim to propose a predictive model for water influx of a reverse osmosis desalination system using multiwalled carbon nanotubes by artificial neural networks (ANNs) [83,84]. Therefore, in order to create, more than 125 numerical cases are simulated using the numerical cases with different boundary conditions. Then, we extracted the data from the simulations and divided the independent data into two categories, training and testing, the ratio of which is 70–30%. Next, by using the hyperparameter tuning, we have optimized the prediction capabilities of the presented models. Additionally, the results are evaluated by using the mean absolute error (MAE) and coefficient of determination ($R^{2}$). The following sections present the optimization of hyperparameters and the results. Figure 9 shows the widely used activation functions.

### 4.1. The Hyperparameter Tuning Process

As was already mentioned, selecting the optimal model to forecast an output parameter is a process that has to be done carefully. Therefore, the hyperparameter tuning of the flux is presented.

In the first step of this process, as mentioned in [61,63], the structure of the hidden layers is investigated. We studied whether increasing the number of hidden layers would significantly affect the predictive capabilities of the output parameters. In Table 2, the structures are analyzed. The simplest structure comprises one hidden layer with 32 neurons, but the most complicated one consists of eight layers with different numbers of neurons.

It is clear that the model improves in accuracy as the number of neurons and hidden layers increases; however, when overfitting occurs, this accuracy decreases and the reverse trend is shown. Consequently, the chosen model is (32,64,128,64,32). Table 3 examines the effects of various activation functions for the output layer after choosing the number of hidden layers.

The batch size, or the amount of data points traveling through the feed-forward process before the backpropagation starts, is another model design component. In other words, batch size refers to the number of data points the neural network processes before updating its weights and biases. As shown in Table 4, 32 is the chosen batch size.

Finally, as seen in Table 5, the number of epochs is contrasted. The model with 20,000 epochs is the ideal option.

The deepest ANN or the slowest model is not always the most excellent choice for hyperparameters. Hence, a search for the optimum model architecture should be conducted before model selection. This selection procedure is carried out to determine the ideal model architecture for each research outcome. Table 6 displays the final chosen models for all sizes of interest. Among the examples examined, these models perform the best.

Therefore, in the following sections, the results of the predictive models are presented.

The flux is predicted using the ANN model, and the results are illustrated in Figure 10. This model inherited the hidden layer structure of (32,64,128,64,32). It is run for 40,000 epochs with a batch size of 8. The input parameters are time, surface roughness, MWCNT-P concentration, and contact angle. The best case is the y = x line where the predicted and numerical results are the same. Moreover, 10% error lines are plotted to give a better sense of the model capabilities.

The results have shown great accuracy. As mentioned previously, 70% of the extracted data from the experimental results are used for training the models. Then, the other 30% is utilized for the evaluation and testing of the model. Figure 10 demonstrates the accuracy of the predictive models, and the mean absolute error is used. Additionally, the $R^{2}$ is used to measure the model’s ability to predict flux. The mentioned parameters are 1.38% and 0.98, respectively.

### 4.2. Predictive Model for Normalized Flux

A similar analysis is done on the normalized flux of the reversed osmosis desalination system. The structure of hidden layers is (64,64,32,32). The batch size is 16, and the model ran for 30,000 epochs. The activation function of the output layer is ReLU.

Figure 11 shows that the model possesses an MAE of 1.02%. Additionally, the $R^{2}$ is equal to 0.99. The results prove that the ANN models could be used to replace the experiments in the mentioned range. The reason behind this is two-fold. First, the accuracy of the predictive models are so good that in some cases they are equal to the experimental results. Second, the experimental expenses are way higher than ANN’s. This is also proved to be correct in [19]. In the present study, the simulation time for the ANN models is almost 3% of the experiments. Therefore, it seems logical to use this model for further utilization of the models to design new thermal systems.

## 5. Conclusions

Synthesized nanocomposites using a modification of carbon nanotubes and an increase in functional groups on the surface of synthesized membranes have shown acceptable performance. Considering the two important factors of the diffusion coefficient and solubility coefficient on the surface of reverse osmosis membranes, materials that can help the hydrophilicity of the membrane surface can increase the flux and salt rejection. In this research, new results were obtained about membrane performance due to the use of new nanocomposites. Among the notable results is the efficiency of more than 97% in salt removal. Additionally, the increase in flux in all nanocomposite membranes with pyrrole was up to 30.8 LMH, which was a 41% improvement compared to plain polyamide membranes. A significant reduction in clogging in nanocomposite membranes containing a pyrrole structure was observed. In relation to the hydrophilicity of the membrane surface due to the presence of hydrogen bonds, the increase in hydrophilicity was associated with an increase in concentration, and also, due to high hydrophilicity, the spread of water particles on the membrane surface increased and caused an increase in flux. According to the SEM analysis, the roughness of the membrane surface due to the increase of N-H groups on the surface of the membrane was reduced compared to plain polyamide membranes. Finally, it can be concluded that the use of nanocomposite membranes with pyrrole, due to the presence of pyrrole monomer on the surface of raw nanotubes, has improved the performance of the membrane, including flux, salt rejection, and clogging, and these membranes can be used as the best nanocomposite membrane. Finally, the proposed model proved to be satisfactorily accurate. Therefore, it could be used to replace further experiments with the mentioned membranes. This would save a lot of time and resources. The proposed model utilizes hyperparameter tuning, which gives the model the advantage of optimized hyperparameters.

## Figures and Tables

**Figure 1 membranes-13-00031-f001:**
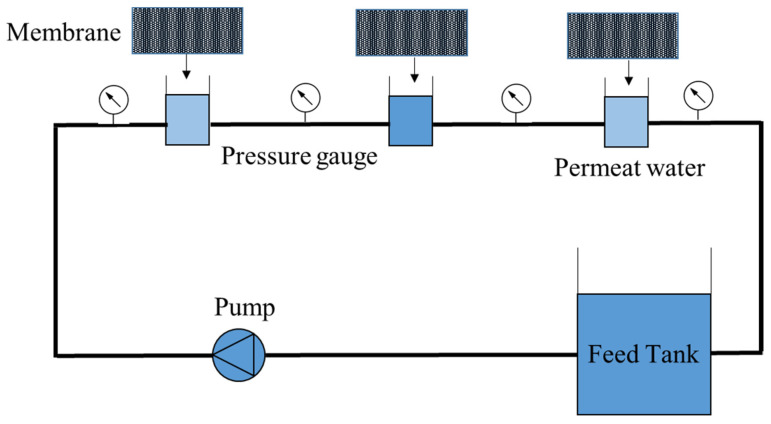
The reverse osmosis model.

**Figure 2 membranes-13-00031-f002:**
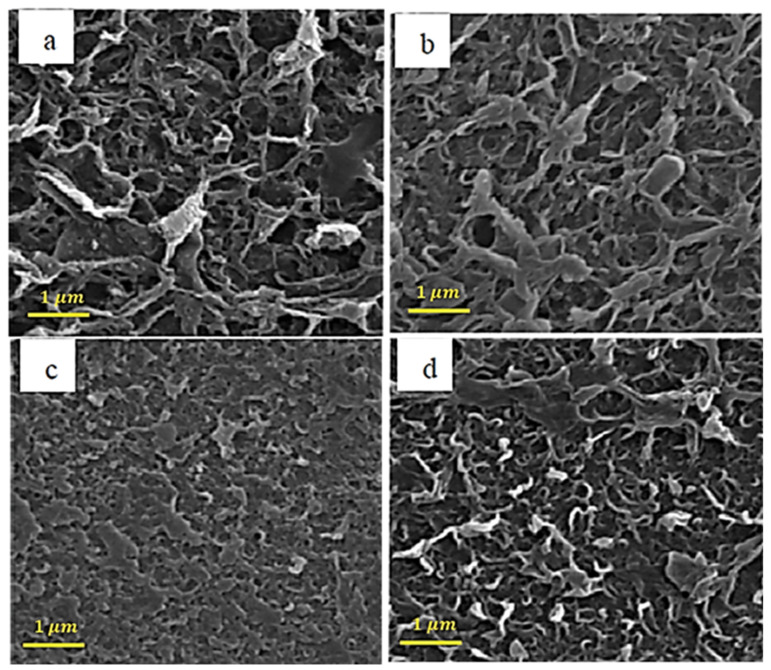
SEM images of the membranes with different loadings of pyrrol (**a**) 0.001%, (**b**) 0.002%, (**c**) 0.005%, and (**d**) 0.01%.

**Figure 3 membranes-13-00031-f003:**
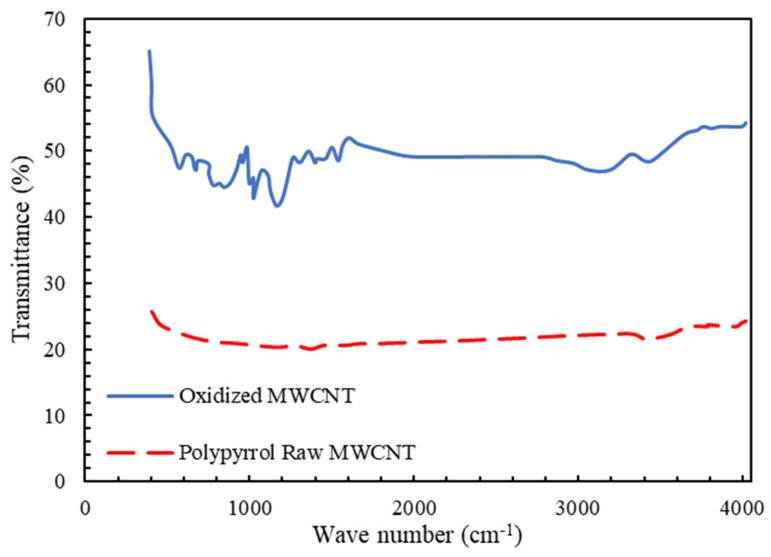
FTIR spectra for determination of compounds on multiwalled carbon nanotubes.

**Figure 4 membranes-13-00031-f004:**
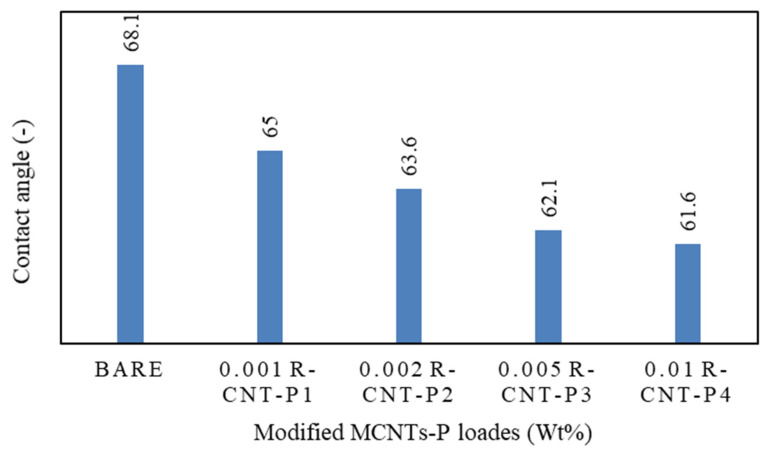
Contact angle analysis of different membranes.

**Figure 5 membranes-13-00031-f005:**
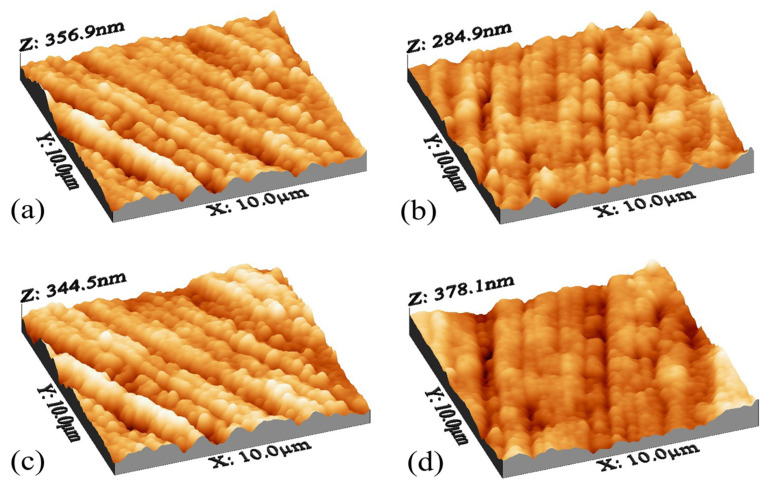
AFM analysis for modified membranes for (**a**) 0.001, (**b**) 0.002, (**c**) 0.005, and (**d**) 0.01.

**Figure 6 membranes-13-00031-f006:**
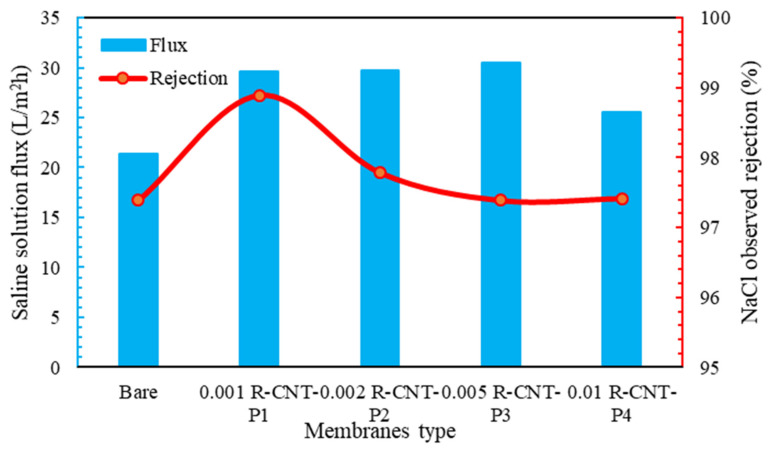
Flux and percent salt rejection for raw membranes.

**Figure 7 membranes-13-00031-f007:**
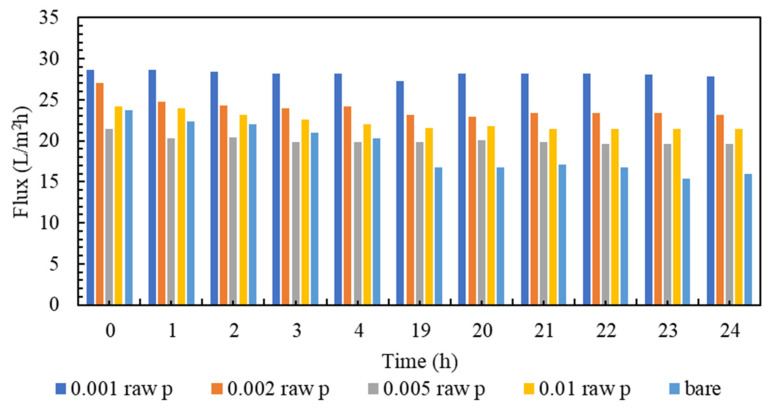
A comparison of Fouling flux of the modified membranes to the bare one.

**Figure 8 membranes-13-00031-f008:**
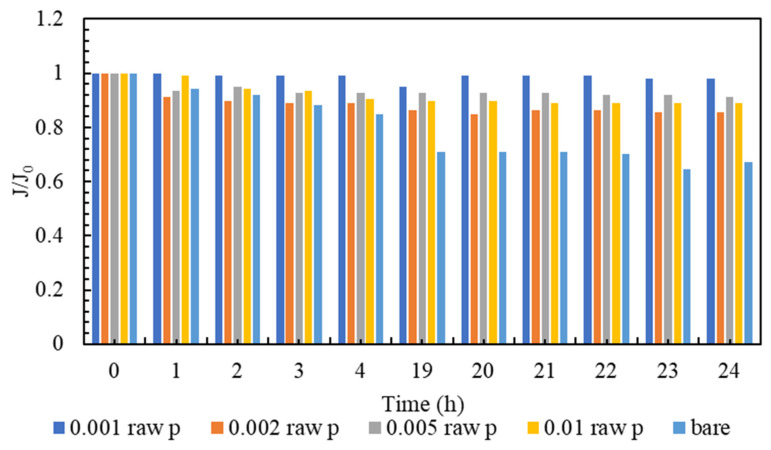
A comparison of Normalized flux (J/J0) of the modified membranes to the bare one.

**Figure 9 membranes-13-00031-f009:**
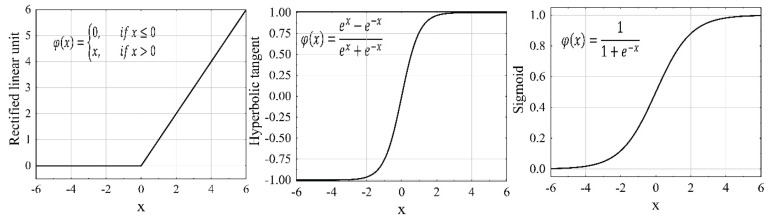
The most well-known activation functions.

**Figure 10 membranes-13-00031-f010:**
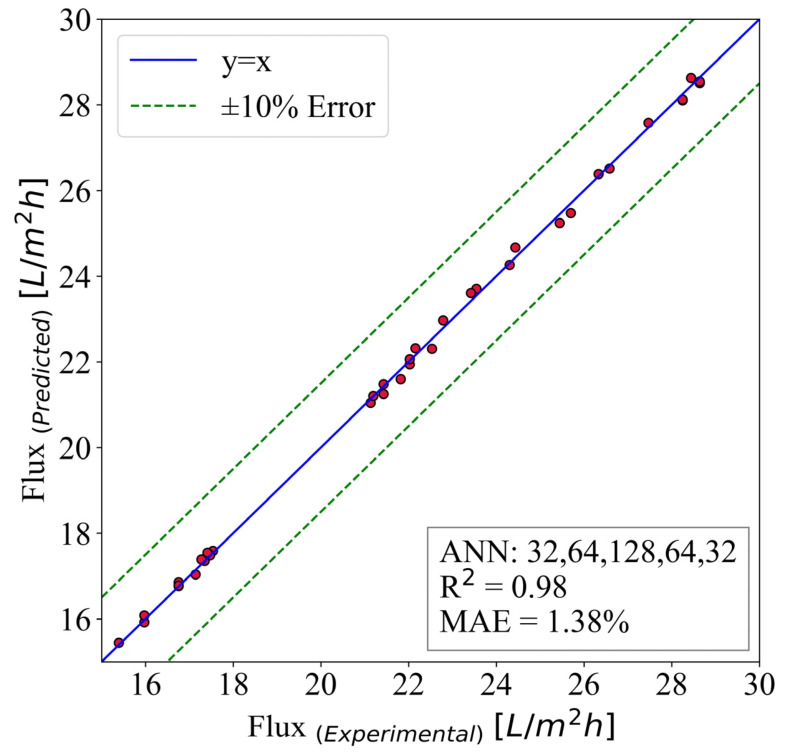
The ANN predictive model for flux.

**Figure 11 membranes-13-00031-f011:**
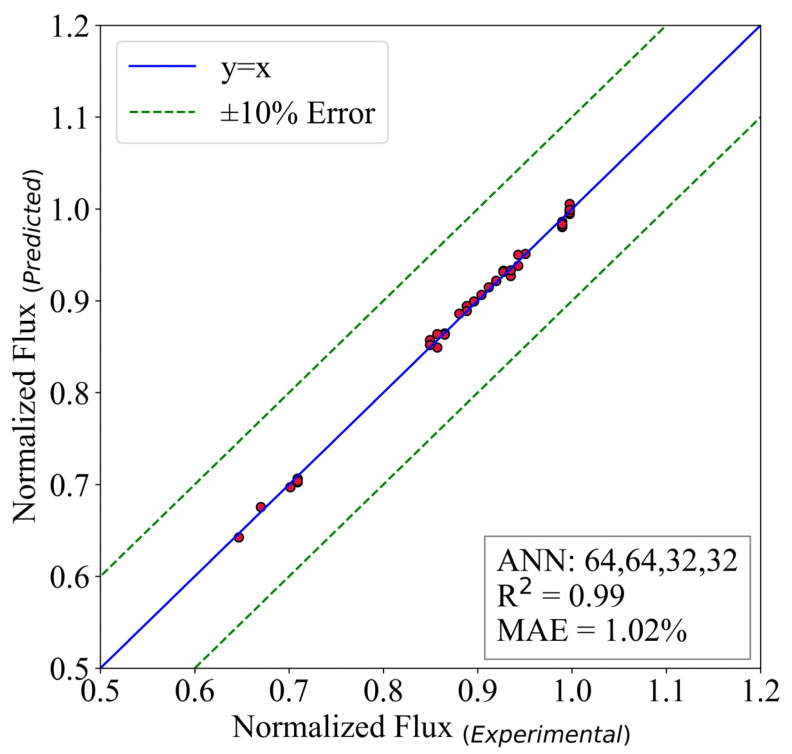
The predictive model for normalized flux.

**Table 1 membranes-13-00031-t001:** The roughness analysis of the surface by DME/APM.

Membrane	Roughness Parameter		
	$S_{a}$ (nm)	$S_{q}$	$S_{v}$
Bare PA RO	54.6	68.8	431
0.001 R-CNT-P1	16.39	16.39	206.8
0.002 R-CNT-P2	32.11	42	304.2
0.005 R-CNT-P3	54.6	67.5	449.4
0.01 R-CNT-P4	28.71	34.98	210.1

**Table 2 membranes-13-00031-t002:** The study of hidden layers formation on the predictions.

Number	Input	Hidden Layers	MAE (%)	$R^{2}$
1	$P,\;G,\;q,\;x_{in},\;\varphi$	(32)	3.98%	0.95
2		(32,64)	2.67%	0.96
3		(32,64,32)	2.61%	0.95
4		(32,64,64,32)	2.48%	0.96
**5 ***		**(32,64,128,64,32)**	**2.06%**	**0.98**
6		(32,64,128,128,64,32)	2.39%	0.98
7		(32,64,128,256,128,64,32)	2.30%	0.97
8		(32,64,128,256,256,128,64,32)	2.41%	0.97
9		(32,64,128,256,512,256,128,64,32)	2.40%	0.97

* Finally selected model.

**Table 3 membranes-13-00031-t003:** Investigation of output layers activation function.

Number	Hidden Layers	Output Activation Function	MAE (%)	$R^{2}$
1	(32,64,128,64,32)	Linear	1.48%	0.98
2		ReLU	2.06%	0.98
3 *		Sigmoid	1.45%	0.99

* Finally selected model.

**Table 4 membranes-13-00031-t004:** The effect of batch size on the results.

Number	Hidden Layers	Batch Size	MAE (%)	$R^{2}$
1	(32,64,128,64,32)	2	1.84%	0.97
2		4	1.46%	0.97
3		8	1.45%	0.97
4		16	1.45%	0.98
5 *		32	1.39%	0.98
6		64	2.47%	0.96

* Finally selected model.

**Table 5 membranes-13-00031-t005:** Study of epochs and their effect on results.

Number	Hidden Layers	Epochs	MAE (%)	$R^{2}$
1	(32,64,128,64,32)	1000	3.81%	0.94
2		2000	2.67%	0.96
3		5000	2.01%	0.97
4		10,000	1.95%	0.98
5 *		20,000	1.36%	0.98
6		30,000	1.39%	0.98
7		40,000	1.38%	0.98
8		50,000	1.38%	0.98

* Finally selected model.

**Table 6 membranes-13-00031-t006:** The proposed models for the output parameters.

Objective Function	Inputs	Hidden Layers	Epochs	Batch Size	Activation Function
Flux	$t,\;S_{a},\;MWCNT-P,\;\varphi \;$	(32,64,128,64,32)	20,000	32	Sigmoid
Normalized flux (J/J0)	$t,\;S_{a},\;MWCNT-P,\;\varphi$	(64,64,32,32)	30,000	16	ReLU

## Data Availability

The data presented in this study are available on request from the corresponding author.

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
