# Peer review of "Study of Membranes with Nanotubes to Enhance Osmosis Desalination Efficiency by Using Machine Learning towards Sustainable Water Management"

_membranes, 2022, doi:10.3390/membranes13010031_

Round 1
Reviewer 1 Report
The manuscript deals with the preparation of membranes with carbon nanotubes to improve processes with membranes. Today it is important to obtain membranes that reduce their most common problems such as clogging and fouling. Therefore, I consider that the manuscript is suitable for publication in this journal. I suggest making the following corrections to improve the manuscript.
Page 3, line 109, to define LMH.
Page 3. line 150, what is SDS?
Page 5. line 228, To check if “… 15 bar in the second year…” is it correct?
Page 5, aquation 1, to center the “J”
Page 6, line 259, To check the percentages, “… (0.001, 0.002, 0.005, and 1.01)…”
Page 6, line 265, To check “…its desalination percentage; At this stage,…”
Page 6, The section is poor in discussion, 3.1. Investigation of synthesized membranes for reverse osmosis process. Please improve the discussion, since it is not specified which figure is referred to when you have the pyrrole
The figure 2 SEM is not clear the descriptions.
Page 7, figure 3, the discussion is not clear, please to point the peaks of wave number inside the figure (687 (N_H) and 3300 (C_H), 1385 and 1540). Please, to change N_H, C_H to N-H, C-H.
Page 8, Table 1, differentiate between Sa, Sq y Sv, in the discussion only mentioned “S”.
Page 9, line 346, To correct superscripts “… flux is 29.5 L/m2h, but at…”
Line 347, To correct superscripts “… 30.8 L/m2h. According…”
Page 10, figure 7, it is not mentioned in the text, if is the same information than the figure 8, to eliminated figure 7.
Page 11, figure 8, If the flux is Normalized, it is dimensionless.
Page 13, Mention that they estimate the heat transfer coefficient, at what is the temperature that you make the estimate?
Author Response
Dear Editor,
We are grateful for the reviewers' comments, their time and consideration. We appreciate direct messages from the reviewers that move forward the text we submitted originally. We have been working following the instructions (in part), and now look forward to meeting your expectations.
We have highlighted the changes within the manuscript, showing the interventions in the text, based on the reviewer’s comments.
I hope we will reach to some extent the high standards of this excellent MDPI journal.
For the authors’
Velibor Spalevic

Reviewer 2 Report
Manuscript Title so long, Minimize with effective title
Author Response

(The authors gave the same response as above.)

Reviewer 3 Report
In generals:
1. In this paper, the improvement in flux, salt rejection and in preventing clogging of osmosis desalination processes by modified membrane material was presented. The membrane material tests were completed. However, there is a conflict argument needs more explanation. The roughness of the membrane is anticipated to disrupt concentration on the membrane surface. A more rougher membrane surface may lead to more flux. But the surface topological and AFM analysis concluded that the rougher membrane leads to less flux. That is contradict to the argument mentioned before.
2. Fat is not the only factor that may clog the membrane surface. Biofouling is the major clogging problem in membrane osmosis system. Author needs to explain why only fat was considered as clogging factor in this study.
3. In the hyperparameter tuning process, the proposed ANN predictive model was used to optimize the parameters. However, the parameters setting for the performance of flux may be contradict to that of clogging. Author needs to explain what objective function was set in the developing of the predictive model.
In Specifics:
1. In page 13 Line 436 and 446: “heat transfer coefficient is predicted…” what is the heat transfer coefficient mentioned in this paper?

Author Response

(The authors gave the same response as above.)
